# Fatty Acid Composition of Muscle and Adipose Tissue in Pigs Fed with Addition of Natural Sorbents

**DOI:** 10.3390/ani12131681

**Published:** 2022-06-29

**Authors:** Piotr Domaradzki, Bożena Nowakowicz-Dębek, Łukasz Wlazło, Mateusz Ossowski, Małgorzata Dmoch, Mariusz Florek

**Affiliations:** 1Department of Quality Assessment and Processing of Animal Products, University of Life Sciences in Lublin, Akademicka 13, 20-950 Lublin, Poland; piotr.domaradzki@up.lublin.pl (P.D.); malgorzata.dmoch@up.lublin.pl (M.D.); 2Department of Animal Hygiene and Environmental Hazards, University of Life Sciences in Lublin, Akademicka 13, 20-950 Lublin, Poland; bozena.nowakowicz@up.lublin.pl (B.N.-D.); lukasz.wlazlo@up.lublin.pl (Ł.W.); mateusz.ossowski@up.lublin.pl (M.O.)

**Keywords:** pig, biochar, montmorillonite, clinoptilolite, muscle tissue, adipose tissue, fatty acids

## Abstract

**Simple Summary:**

Different types of unconventional feed resources as feed additives for animals are used worldwide on the basis of their availability and economical consideration. Natural mineral sorbents have been used in a wide range in animal farming mainly as feed additives since the mid-1960s. Still, to date, no studies have investigated the effect of the addition of natural sorbents such as biochar, bentonite–montmorillonite and zeolite–clinoptilolite to feed on the fatty acid content of pig meat (muscle tissue) and fat (adipose tissue). The presented research carried out in this area has revealed some evidence of fatty acid profile modification. In spite of the fact that effects of investigated sorbents were often opposite, they did not adversely affect indices of nutritional and pro-health quality of adipose tissue of the pig during fattening. Due to the fact that effects of sorbents supplementation are highly variable and can depend on the composition and dosage, further research seems fully reasonable.

**Abstract:**

The fatty acid composition of meat and fat was studied in Choice Genetics line pigs fed a diet with three natural sorbents. Control (C1 and C2), biochar (D, 0.5%), bentonite–montmorillonite (A, 1.5%) and zeolite–clinoptilolite (B, 1.5%) diets were used in two trials. The samples of back fat, kidney fat and Longissimus lumborum (MLL) and Semimembranosus (MSM) muscle were examined. All sorbents (D, A and B) had no effect on fatty acid composition in MLL, whereas in MSM turned out to be very limited and inconsistent. Although A and B sorbents had a significant impact on the fatty acid profile of kidney fat, their effect was often opposite. Sorbent B’s effects were less beneficial due to a significantly higher proportion of saturated fatty acids, higher value of thrombogenic and atherogenic indexes, *n*-6/*n*-3 ratio but lower h/H ratio. Sorbent A’s effects significantly increased polyunsaturated fatty acids, and positively influenced lipid health quality indices. In summary, the feeding of natural sorbents slightly modified the fatty acid profile of muscle tissue, kidney fat and back fat; however, it did not have a negative effect on the indices of nutritional and pro-health quality of adipose tissue of pigs during fattening.

## 1. Introduction

The OECD-FAO estimates by 2025 an increase of meat production by 1.8 Mt annually, primarily in pork and poultry products [1]. The changes of global dynamics of the production and consumption of meat and meat products nowadays include different issues, in particular nutritional, safety and sustainability challenges. Among these calls for action, it is particularly important to optimise the content and profile of lipid in meat and meat products, taking into account health recommendations [2]. The goal of animal meat production is to increase efficiency of production, maximize muscle growth, minimize fat deposition and produce high-quality animal tissues rich in bioactive components. Determinants of animal carcass composition (meat and fat raw materials) are very complex and include inter alia species, breed, age, sex, feeding type, etc. Among production practices, nutrition and feeding management represent the first opportunity to modify lipids (content of fat, composition of fatty acid, and level of cholesterol) [3]. From the point of view of the economics of pig production, with few exceptions, the excessive fat deposition is wasteful. However, the intramuscular fat content should be satisfactory to provide the desirable sensory attributes of high-quality meat products [4].

Feed additives are used in feed for animals in order to achieve a favourable change in the feed, on the animal’s organism, on food products of animal origin and on the environment [5]. Papaioannou et al. [6] suggested that the propagation of sorbents use in animals diet apart from the improvement of health status, in addition contribute to a potential enhancement in the final quality of animal products. All around the world, farmers use various categories of unconventional feed resources as feed additives, taking into account their availability and economic rationale [7]. Mineral sorbents are used in swine farming, mostly as feed additives for animals at different ages: biochar (charcoal) within the limits of 0.3 to 0.6% [8], zeolites in the amount of 0.5% to 8% [7,9,10] and from 0.3% to 2% in the case of bentonites [11,12,13]. In the case of synthetic zeolites (sodium aluminosilicate), there are insufficient data on both the safety and the efficacy of their use in animal nutrition [14]. In the European Community, all legal aspects relating to the use of feed additives, including primarily the procedure for authorizing and the placing on the market and labelling is established by Regulation No. 1831/20034 [15].

Previously, authors examined the effect of three natural sorbents in the diet of fattening pigs on two skeletal muscles and demonstrated that the sorbents have no negative effect on the physicochemical properties of meat and use it as case-ready meat or processing material [16]. To the best of our knowledge, to date, no other studies have described the effect of the addition of natural sorbents to feed on the fatty acid content of pig meat and fat. Therefore, the aim of the present research was to investigate the effect of feeding natural sorbents (biochar, bentonite–montmorillonite and zeolite–clinoptilolite) as a supplement for the diet of crossbred fattening pigs on fatty acid content in two skeletal muscles and two adipose tissues.

## 2. Materials and Methods

### 2.1. Ethical Approval

The study was approved by the Local Ethics Committee on Animal Experimentation of University of Life Sciences in Lublin, Poland (approval no. 100/2015, of 8 December 2015), and the experiment was conducted according to the guidelines of the Declaration of Helsinki and in compliance with the European Union law (Directive 2010/63/UE, received in Poland by Legislative Decree 266/2015) of the European Parliament and of the Council on the protection of animals used for scientific or educational purposes.

### 2.2. Experimental Design, Animals, Diet and Sampling

This study supplements the information provided by Ossowski et al. [16] where the fattening pigs (multi-breed crosses of the Choice Genetics line), housing, treatment and diet has been fully described. The investigations were performed under the project “Development of innovative technologies of comprehensive utilization of waste generated during pig fattening-Computil”. In this experiment, the effectiveness of a feed additive containing powdered aluminosilicates (bentonite and zeolite) on pig productivity was investigated. The ready-to-use feed additive as well as the method of its production are the subject of the patent application (The Patent Office of the Republic of Poland, no. P.432877). Biochar (0.5%) was used as a third sorbent. The experiments (no. 1 and 2) were conducted on a single farm in two consecutive production cycles. Supplementary information about the ingredients of the diets can be retrieved from Ossowski et al. [16]. Experiment 1 was completed in June and the Experiment 2 in December. Detailed information about sampling can be found in Ossowski et al. [16]. Briefly, six fatteners (slaughter weight between 105–108 kg) from each group (three male and three female) were randomly selected for slaughter, then transported to the slaughter plant and killed in accordance with Council Regulation (EC) no. 1/2005 of 22 December 2004 [17] and Council Regulation (EC) no. 1099/2009 of 24 September 2009 [18], respectively.

The research material comprised samples of two adipose tissues: subcutaneous (back fat) and internal (kidney fat), and two skeletal muscles: longissimus lumborum (MLL) and semimembranosus (MSM). Representative samples of tissues were collected during carcass fabrication after 24 h chill time, then separately vacuum-packed in PA/PE bags with a high gas barrier and a 98% vacuum level and stored at −45 °C (LT U250, Nordic Lab., Vaerloese, Denmark) until analysis within one month.

### 2.3. Fatty Acid Analysis

The fatty acid (FAs) profile of lipids from adipose and muscle tissues was determined following fat extraction according to method of Folch et al. [19]. Methyl esters of FAs (FAME) were prepared by the transmethylation of fat samples (50 mg) using a mixture of concentrated H_2_SO_4_ (95%) and methanol according to the AOCS Official Method Ce 2-66 [20]. Gas chromatographic (GC) analyses were performed in detail according to Domaradzki et al. [21]. The fatty acid composition was expressed as percentage of total identified FAs. The analyses were carried out in duplicate. The following groups of fatty acids were identified: saturated FAs (SFA: sum of C8:0, C10:0, C12:0, C14:0, C15:0, C16:0, C17:0, C18:0, and C20:0); *cis* monounsaturated FAs (MUFA *cis*: sum of C14:1 *n*-5, C15:1 *n*-6, C16:1 *n*-9 C16:1 *n*-7, C17:1 *n*-8, C18:1 *n*-9, C18:1 *n*-7, C18:1 *n*-5, C20:1 *n*-11, and C20:1 *n*-9) and polyunsaturated FAs (PUFA: sum of C18:3 *n*-3, C20:3 *n*-3 C20:5 *n*-3 C22:5 *n*-3 C22:6 *n*-3, C18:2 *n*-6, C18:3 *n*-6, C20:2 *n*-6, C20:3 *n*-6, C20:4 *n*-6, C22:4 *n*-6, and C20:2 *n*-9);. In the tables, the 10 most important FAs with the highest percentage in each identified group of FAs were presented. In addition, the following ratios and indices were calculated: PUFA *n*-6/*n*-3, PUFA/SFA, h/H—hypocholesterolaemic/hypercholesterolaemic ratio: sum of [(C18:1 *n*-9, C18:2 *n*-6, C18:3 *n*-3, C18:3 *n*-6, C20:2 *n*-6, C20:3 *n*-6, C20:4 *n*-6, C20:3 *n*-3, C20:4 *n*-3, C20:5 *n*-3, C22:4 *n*-6, C22:5 *n*-6, C22:5 *n*-3 and C22:6 *n*-3)/(sum of C12:0, C14:0 and C16:0)]; AI—atherogenic index [(sum of C12:0, 4 × C14:0 and C16:0)/(sum of MUFA *cis*, PUFA *n*-6 and *n*-3)] [21], and TI—thrombogenic index [(sum of C14:0, C16:0 and C18:0)/(sum of 0.5 × MUFA *cis*, 0.5 × PUFA *n*-6, 3 × PUFA *n*-3 and PUFA *n*-3/*n*-6)] [22].

### 2.4. Statistical Analysis

The statistical analyses were performed by Statistica ver. 13 (TIBCO Software Inc., Palo Alto, CA, USA). A general linear model for one-way ANOVA analysis was used to estimate the fixed effect of the feeding group. The fatty acid composition in tissues were calculated for six pigs per group, using the pen as a replicate (*n* = 3, per group), and the pen was considered the average value for two pigs taken for analysis from it. The significance of differences between means for groups was determined by the unpaired Student’s *t*-test (for two independent groups), and Tukey’s HSD test (for multiple comparisons in Experiment 2). Statistical significance was set at *p* < 0.01 or *p* < 0.05, whereas *p* < 0.10 was considered a tendency. The results were expressed as the mean value and standard error of the mean.

## 3. Results

Table 1 shows the percentage of fatty acids in the intramuscular fat (IMF) of the evaluated skeletal muscles of fattening pigs from Experiment 1. The addition of biochar (Sorbent D) in the diets did not significantly affect the proportion of all fatty acids in the MLL (except for arachidonic FA, C20:4 *n*-6 AA), and no differences were found in the values of indices and proportions. A greater differentiation was shown in the case of MSM of fatteners fed with biochar, which contained a significantly (*p* < 0.01) higher share of stearic acid (C18:0) (by 0.92 p.p.), whereas significantly (*p* < 0.01) less C16:1 *n*-7 (by 0.36 p.p.) and C18:1 *n*-9 acids (by 2.74 p.p.) and ΣMUFA *cis* (by 3.15 p.p.) was found, in comparison with the control group. Similar to MLL, a significantly higher proportion of arachidonic acid was also found (by 0.69 p.p., *p* < 0.05) in fatteners fed with biochar.

Table 2 presents results of the percentage of fatty acids in the kidney fat and back fat of fattening pigs from Experiment 1. The percentage of fatty acids as well as their sums, indices and proportions in backfat of pigs receiving the biochar supplement did not differ significantly between feeding groups, except for the lower content of docosahexaenoic acid (C22:6 *n*-3, DHA). Greater variation was observed for kidney fat. Fatteners fed with biochar supplement showed a significantly (*p* < 0.01) higher content of stearic acid (by 1.49 p.p.), lower content of oleic acid (by 2.12 p.p.) and a lower ΣMUFA *cis* (by 2.49 p.p.) in comparison with the control group, for which a significantly (*p* < 0.01) lower proportion of *n*-6/*n*-3 acids (7.66 vs. 8.24) and a lower proportion of arachidonic acid (*p* < 0.10) were found.

The fatty acid profile in the intramuscular fat of MLL and MSM, and in the kidney fat of the fattening pigs from Experiment 2 is given in Table 3 and Table 4, respectively. The addition of Sorbents A (with montmorillonite prevalence) and B (with clinoptilolite prevalence) had no effect on the proportion of fatty acids in the muscle tissue from MLL (Table 3), and there were not even any trends.

In the case of MSM (Table 3), the addition of sorbents modified the fatty acid profile to a certain but limited extent, as indicated by the differences and trends found. However, it is difficult to clearly indicate whether the direction of the observed changes is beneficial from the consumer’s point of view. The addition of Sorbent A (montmorillonite) influenced the highest proportion of oleic acid (C18:1 *n*-9, *p* < 0.05) and ΣMUFA *cis* (*p* < 0.10) in comparison with C2 and B groups, and at the same time, the lowest value of *n*-6/*n*-3 ratio was observed (*p* < 0.01). The unfavourable trends shown in the use of both sorbents include a higher proportion of ΣSFA but lower ΣPUFA, C20:4 *n*-6 AA, C20:5 *n*-3 EPA and a lower value of PUFA/SFA ratio in the intramuscular fat of MSM.

The use of sorbents had a significant effect on the fatty acid profile of kidney fat, with the effect of sorbents often being opposite (Table 4). Sorbent B (clinoptilolite) affected a higher proportion of saturated fatty acids (ΣSFA, *p* < 0.05 and C16:0, *p* < 0.01) and a higher value of the *n*-6/*n*-3 ratio (*p* < 0.05). Compared to the C2 group, the Sorbent A (montmorillonite) significantly affected the increase in proportion of ΣPUFA (by 3.47 p.p., *p* < 0.05), Σ*n*-3 (by 0.44 p.p., *p* < 0.05), and C18:3 *n*-3 ALA (*p* < 0.05) and C22:6 *n*-3 DHA (*p* < 0.01). Such positive trends (*p* < 0.10) were also observed for Σ*n*-6, C18:2 *n*-6 LA and C20:4 *n*-6 AA. In contrast, Sorbent B (clinoptilolite) showed a much weaker effect (than Sorbent A) and the differences with the control group were not significant.

The positive effect of Sorbent A (montmorillonite) was also confirmed in terms of lipid health quality indices, as it influenced, in comparison to the C2 group, a higher PUFA/SFA ratio (*p* < 0.05) and a lower TI index, whereas AI and h/H indices in both groups were at the same level. On the other hand, the effect of Sorbent B (clinoptilolite) turned out to be less beneficial due to a significant (*p* < 0.05) worsening of the discussed indices, i.e., increase in TI and AI values and decrease in h/H ratios.

## 4. Discussion

Dietary supplementation of wood charcoal increased fat excretion by broilers and decreased fat digestion, reducing available energy [23]. Moreover, the reduction of abdominal fat weight [23] and fat content in broiler carcasses was observed [23,24]. The numerically lower fat content in skeletal muscles of fattening pigs supplemented with biochar was also confirmed by the authors in previously published studies [16]. In contrary, opposite results regarding fat content are reported by Chu et al. [8] for longissimus dorsi of fattening pigs fed with the addition of bamboo charcoal.

Many authors observed the decrease of saturated fatty acid and concomitant increase of unsaturated fatty acids (ΣUFA) in meat from broiler chickens fed with bamboo charcoal [24] or activated charcoal [25]. In general, in both groups of fatty acids (SFA and UFA), significant differences were mainly related to fatty acids with 18 carbon atoms (C18:0, C18:1, C18:2 and C18:3). Similar relationships regarding the effect of bamboo charcoal supplementation on the fatty acid profile of the longissimus dorsi muscle of fattening pigs were obtained by Chu et al. [8]. These significant changes (*p* < 0.05) affected both the relationship between ΣSFA and ΣUFA (as a consequence UFA/SFA ratio) and were also associated with an increase in the proportion of 18-carbon unsaturated fatty acids as oleic and linoleic acids; however, the stearic acid and arachidonic acid percentage was significantly decreased. In turn, the charcoal supplementation in a diet was of limited influence on the fatty acid composition (%) of duck breast muscle, however, the reduction of the eicosapantenoic acid (C20:5 *n*-3) and a total content of *n*-3 FAs (*p* < 0.01), but an increase of *n*-6/*n*-3 ratio (*p* < 0.02) were found [26]. Similar tendencies were observed in the present study for muscle tissues (Table 1), considering in particular individual long chain PUFAs, notably C20:4 *n*-6 AA (*p* < 0.05), as well ΣPUFA and Σ*n*-6 (MSM, *p* < 0.10). However, according to FAO/WHO experts [27], there is no rational for a specific recommendation for *n*-6/*n*-3 ratio (or LA/ALA ratio) if intakes of both *n*-6 and *n*-3 FAs fulfill the recommended amounts (essential LA 2.5–3% of energy and essential ALA >0.5% of energy).

The comparison of the present results with those from the literature is on the one hand difficult, and on the other hand not really possible due to the lack of similar studies conducted on pork and porcine fat from pigs supplemented with sorbents. Clay minerals have been reported to improve nutrient digestibility and enzymatic activity in broilers [28,29], to maintain the intestinal integrity in weaned piglets [30], which may contribute to enhanced nutrient utilization. However, the effectiveness of sorbents is dose dependent, and it was found for broiler chickens and growing swine that greater supplementation is less effective than lower levels [10,31]. Moreover, Lv et al. [32] found that at lower doses, palygorskite (magnesium aluminum silicate) has shown positive effects on performance and nutrient utilization in weaned piglets; however, at higher doses it can absorb nutrients and restrict or impede a nutrient digestibility.

On the one hand, zeolites ingested with the diet lead to the shift of pH and buffering capacity of gastrointestinal secretions and affect the transport through the intestinal epithelium, composition of intestinal bacteria and resorption of bacterial products, vitamins and microelements [33]. On the other hand, bentonites are effective mainly to retain some compounds containing specific chemical groups, particularly those with acidic groups (organic acids) or with multiple-bonds (molecules containing double bonds), as well conjugated compounds [34]. In monogastric animals (vs. ruminants), some fatty acids present in the feed are absorbed and remain unchanged in the intestines [35]. In pigs, saturated and monounsaturated fatty acids are synthesized in vivo, thus are less readily influenced by diet than the polyunsaturated fatty acids. For example, LA (C18:2 *n*-6) and ALA (C18:3 *n*-3) cannot be synthesized; therefore, their concentrations depend on the feeding diet [36]. The present study showed that supplementing a pig diet with different sorbents such as biochar, bentonite (montmorillonite) and zeolite (clinoptilolite) can influence fatty acid composition of muscle and adipose tissues; however, the obtained results are not conclusive. Therefore, different, sometimes contradictory effects, were observed for the effect of the use of the evaluated sorbents on the fatty acid profile of both skeletal muscle tissue and adipose tissue. Furthermore, the results indicate that intramuscular fat from the semimembranosus muscle is more susceptible to nutritional modification of the fatty acid profile than the longissimus lumborum muscle, which can be related to characteristics of muscle fibre types [37]. According to Essén-Gustavsson et al. [38], lipids are stored primarily in the fibres of type I and to a lesser extent in IIA ones. However, Larzul et al. [39] did not find any relationship between IMF content and fibre type composition in purebred large white pigs.

The AI and TI indices are regarded as indicators of the influence of diet on the incidence of coronary heart disease [22]. In turn, the hypocholesterolaemic/hypercholesterolaemic fatty acids ratio (h/H) provides a more objective indices of the nutritional evaluation of fat, since some SFAs (for example C18:0) do not influence plasma cholesterol and increase the beneficial effects of MUFA [40]. In terms of the above three indices, the fat from both muscle tissues and adipose tissues of pigs fed diet with biochar supplementation (Sorbent D, Experiment 1) had similar values compared to control group, i.e., in the end, its influence has been a neutral one. Despite evaluating the effect of Sorbents A (montmorillonite) and B (clinoptilolite) in Experiment 2 on health-promoting indices of fatty acids of the meat and fat of pigs, a more favourable effect was observed for montmorillonite (bentonite) than for clinoptilolite (zeolite).

## 5. Conclusions

Overall, the results indicated that feeding of natural sorbents rather slightly modified the fatty acid profile of muscle tissue, kidney fat and back fat, as well did not have a negative effect on the indices of nutritional and pro-health quality of adipose tissue from pigs during fattening. However, the presented results should be treated as preliminary. Thus, further research seems fully reasonable.

## Figures and Tables

**Table 1 animals-12-01681-t001:** Composition of fatty acid (% of total fatty acids) of Musculus longissimus lumborum and Musculus semimembranosus.

Fatty Acid (%) and Indices	Musculus Longissimus Lumborum				Musculus Semimembranosus			
	Control (1)	Sorbent D	SEM	Significance	Control (1)	Sorbent D	SEM	Significance
16:0	25.61	25.73	0.25	ns	23.00	23.37	0.17	ns
18:0	12.43	12.83	0.11	<0.10	10.45 ^A^	11.37 ^B^	0.20	<0.01
∑SFA	40.03	40.45	0.31	ns	35.25	36.52	0.35	<0.10
16:1 *n*-7	3.39	3.38	0.10	ns	3.67 ^B^	3.31 ^A^	0.07	<0.01
18:1 *n*-9	37.81	37.69	0.20	ns	37.78 ^B^	35.04 ^A^	0.53	<0.01
∑MUFA *cis*	47.27	47.06	0.26	ns	48.96 ^B^	45.81 ^A^	0.53	<0.01
18:2 *n*-6 LA	9.14	8.58	0.36	ns	10.75	11.66	0.32	ns
18:3 *n*-3 ALA	0.61	0.51	0.03	ns	0.59	0.56	0.01	ns
20:4 *n*-6 AA	1.15 ^a^	1.45 ^b^	0.07	<0.05	2.00 ^a^	2.69 ^b^	0.16	<0.05
20:5 *n*-3 EPA	0.17	0.19	0.01	ns	0.26	0.32	0.02	ns
22:5 *n*-3 DPA	0.33	0.39	0.02	ns	0.47	0.58	0.03	ns
22:6 *n*-3 DHA	0.23	0.26	0.01	ns	0.30	0.37	0.02	ns
∑PUFA	12.46	12.25	0.48	ns	15.51	17.40	0.58	<0.10
∑*n*-3	1.41	1.42	0.07	ns	1.71	1.89	0.07	ns
∑*n*-6	10.95	10.72	0.42	ns	13.67	15.37	0.51	<0.10
*n*-6/*n*-3	7.83	7.62	0.14	ns	7.98	8.17	0.08	ns
PUFA/SFA	0.31	0.30	0.01	ns	0.44	0.48	0.02	ns
TI	1.18	1.20	0.02	ns	0.95	0.99	0.02	ns
AI	0.43	0.44	0.01	ns	0.36	0.37	0.01	ns
h/H	1.85	1.83	0.03	ns	2.18	2.11	0.02	ns

SEM—standard error of mean; *n* = 12; Sorbent D, Biochar 0.5%; h/H, hypocholesterolaemic/hypercholesterolaemic ratio; AI, atherogenic index; TI, thrombogenic index; Means in the same row within muscle with different superscripts are significantly different: ns, not significant; ^a, b^—*p* < 0.05; ^A, B^—*p* < 0.01; tendency—*p* < 0.10.

**Table 2 animals-12-01681-t002:** Composition of fatty acid (% of total fatty acids) of kidney fat and back fat.

Fatty Acid (%) and Indices	Kidney Fat				Back Fat			
	Control (1)	Sorbent D	SEM	Significance	Control (1)	Sorbent D	SEM	Significance
16:0	29.69	29.90	0.41	ns	26.88	26.36	0.49	ns
18:0	16.84 ^A^	18.33 ^B^	0.33	<0.01	14.87	14.82	0.30	ns
∑SFA	48.80	50.48	0.61	ns	43.92	43.36	0.67	ns
16:1 *n*-7	2.26	2.12	0.07	ns	2.31	2.07	0.07	<0.10
18:1 *n*-9	33.19 ^B^	31.07 ^A^	0.46	<0.01	35.61	35.97	0.28	ns
∑MUFA *cis*	38.61 ^b^	36.12 ^a^	0.56	<0.05	41.54	41.67	0.33	ns
18:2 *n*-6 LA	10.37	11.21	0.38	ns	11.84	12.20	0.43	ns
18:3 *n*-3 ALA	1.00	0.99	0.03	ns	1.13	1.12	0.02	ns
20:4 *n*-6 AA	0.12	0.15	0.01	<0.10	0.15	0.15	0.01	ns
20:5 *n*-3 EPA	0.04	0.04	0.00	ns	0.05	0.05	0.00	ns
22:5 *n*-3 DPA	0.14	0.14	0.01	ns	0.17	0.17	0.01	ns
22:6 *n*-3 DHA	0.14	0.14	0.01	ns	0.16 ^B^	0.14 ^A^	0.01	<0.01
∑PUFA	12.34	13.18	0.44	ns	14.22	14.59	0.50	ns
∑*n*-3	1.42	1.43	0.04	ns	1.66	1.63	0.04	ns
∑*n*-6	10.92	11.76	0.40	ns	12.55	12.96	0.46	ns
*n*-6/*n*-3	7.66 ^A^	8.24 ^B^	0.12	<0.01	7.53	7.95	0.15	ns
PUFA/SFA	0.25	0.26	0.01	ns	0.32	0.34	0.02	ns
TI	1.65	1.76	0.04	ns	1.34	1.32	0.04	ns
AI	0.59	0.61	0.01	ns	0.49	0.47	0.01	ns
h/H	1.45	1.40	0.04	ns	1.76	1.82	0.06	ns

SEM—standard error of mean; *n* = 12; Sorbent D, Biochar 0.5%; h/H, hypocholesterolaemic/hypercholesterolaemic ratio; AI, atherogenic index; TI, thrombogenic index; Means in the same row within muscle with different superscripts are significantly different: ns, not significant; ^a, b^—*p* < 0.05; ^A, B^—*p* < 0.01; tendency—*p* < 0.10.

**Table 3 animals-12-01681-t003:** Composition of fatty acid (% of total fatty acids) of Musculus longissimus lumborum and Musculus semimembranosus.

Fatty Acid (%) and Indices	Musculus Longissimus Lumborum					Musculus Semimembranosus				
	Control (2)	Sorbent A	Sorbent B	SEM	Sig.	Control (2)	Sorbent A	Sorbent B	SEM	Sig.
16:0	24.48	24.73	24.98	0.30	ns	23.06	23.62	24.73	0.34	ns
18:0	11.22	11.70	11.48	0.18	ns	10.27	10.74	11.32	0.27	ns
∑SFA	37.71	38.34	38.31	0.41	ns	35.21	36.23	37.89	0.60	<0.10
16:1 *n*-7	3.56	3.46	3.24	0.13	ns	3.61	3.65	3.25	0.11	ns
18:1 *n*-9	37.61	36.28	37.45	0.88	ns	38.89 ^a^	42.15 ^b^	38.96 ^a^	0.63	<0.05
∑MUFA *cis*	48.72	47.15	47.99	0.89	ns	50.19	52.40	49.31	0.63	<0.10
18:2 *n*-6 LA	9.17	9.27	9.83	0.69	ns	9.92	7.88	9.30	0.46	<0.10
18:3 *n*-3 ALA	0.46	0.43	0.44	0.03	ns	0.58	0.54	0.48	0.03	ns
20:4 *n*-6 AA	1.80	1.89	1.79	0.18	ns	1.86	1.19	1.40	0.13	<0.10
20:5 *n*-3 EPA	0.21	0.23	0.18	0.03	ns	0.21	0.15	0.12	0.02	<0.10
22:5 *n*-3 DPA	0.41	0.40	0.43	0.05	ns	0.41	0.28	0.28	0.03	ns
22:6 *n*-3 DHA	0.16	0.29	0.18	0.04	ns	0.18	0.20	0.11	0.02	ns
∑PUFA	13.27	14.19	13.34	1.07	ns	14.28	11.13	12.63	0.69	<0.10
∑*n*-3	1.32	1.40	1.28	0.14	ns	1.47	1.26	1.07	0.09	ns
∑*n*-6	11.77	12.64	11.93	0.94	ns	12.64	9.75	11.46	0.62	<0.10
*n*-6/*n*-3	8.94	9.76	9.24	0.38	ns	8.60 ^AB^	7.77 ^A^	11.01 ^B^	0.58	<0.01
PUFA/SFA	0.35	0.38	0.35	0.03	ns	0.41	0.31	0.36	0.02	<0.10
TI	1.08	1.11	1.11	0.03	ns	0.96	1.02	1.11	0.03	ns
AI	0.40	0.41	0.41	0.01	ns	0.36	0.38	0.40	0.01	ns
h/H	1.95	1.93	1.92	0.04	ns	2.16	2.12	1.98	0.04	ns

SEM—standard error of mean; *n* = 18; A and B sorbent groups receiving feed with 1.5% of mixtures (in different proportions) of bentonite (montmorillonite) and zeolite (clinoptilolite); Sorbent A: bentonite (montmorillonite) > zeolite (clinoptilolite); Sorbent B: zeolite (clinoptilolite) > bentonite (montmorillonite); h/H, hypocholesterolaemic/hypercholesterolaemic ratio; AI, atherogenic index; TI, thrombogenic index; Means in the same row with different superscripts are significantly different: Sig., Significance; ns, not significant; ^a, b^—*p* < 0.05; ^A, B^—*p* < 0.01; tendency—*p* < 0.10.

**Table 4 animals-12-01681-t004:** Composition of fatty acid (% of total fatty acids) of kidney fat.

Fatty Acid (%) and Indices	Control (2)	Sorbent A	Sorbent B	SEM	Significance
16:0	29.55 ^A^	29.63 ^A^	31.26 ^B^	0.30	<0.01
18:0	21.32	20.53	21.87	0.37	ns
∑SFA	52.96 ^a^	52.51 ^a^	55.53 ^b^	0.64	<0.05
16:1 *n*-7	1.37	1.43	1.49	0.04	ns
18:1 *n*-9	32.10	29.08	28.66	0.63	<0.10
∑MUFA *cis*	36.10	33.12	32.59	0.66	ns
18:2 *n*-6 LA	9.06	11.92	9.99	0.56	<0.10
18:3 *n*-3 ALA	0.88 ^a^	1.18 ^b^	0.94 ^ab^	0.06	<0.05
20:4 *n*-6 AA	0.10	0.14	0.11	0.01	<0.10
20:5 *n*-3 EPA	0.03	0.05	0.02	0.01	ns
22:5 *n*-3 DPA	0.11	0.14	0.11	0.01	ns
22:6 *n*-3 DHA	0.07 ^A^	0.13 ^B^	0.07 ^A^	0.01	<0.01
∑PUFA	10.76 ^a^	14.23 ^b^	11.74 ^ab^	0.68	<0.05
∑*n*-3	1.18 ^a^	1.62 ^b^	1.22 ^a^	0.08	<0.05
∑*n*-6	9.58	12.61	10.52	0.60	<0.10
*n*-6/*n*-3	8.13 ^ab^	7.77 ^a^	8.62 ^b^	0.16	<0.05
PUFA/SFA	0.20 ^a^	0.27 ^b^	0.21 ^ab^	0.01	<0.05
TI	1.97 ^a^	1.86 ^a^	2.16 ^b^	0.06	<0.05
AI	0.63 ^a^	0.63 ^a^	0.71 ^b^	0.02	<0.05
h/H	1.38 ^b^	1.38 ^b^	1.22 ^a^	0.03	<0.05

SEM—standard error of mean; *n* = 18; A and B sorbent groups receiving feed with 1.5% of mixtures (in different proportions) of bentonite (montmorillonite) and zeolite (clinoptilolite); Sorbent A: bentonite (montmorillonite) > zeolite (clinoptilolite); Sorbent B: zeolite (clinoptilolite) > bentonite (montmorillonite); h/H, hypocholesterolaemic/hypercholesterolaemic ratio; AI, atherogenic index; TI, thrombogenic index; Means in the same row with different superscripts are significantly different: ns, not significant; ^a, b^—*p* < 0.05; ^A, B^—*p* < 0.01; tendency—*p* < 0.10.

## Data Availability

Data sharing not applicable.

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
