# Peer review of "Fatty Acid Composition of Muscle and Adipose Tissue in Pigs Fed with Addition of Natural Sorbents"

_animals, 2022, doi:10.3390/ani12131681_

Round 1
Reviewer 1 Report
Dear Authors,
generally your manuscript is not too bad but in my opinion you should rewrite it or just introduce some changes in the text.
First you have to decide what was the aim of your study. In the reviewed manuscript I found results' description of skeletal muscles and adipose tissues laboratory analyses - samples (n=35) were obtained from the nutritional (?) experiment and that's ok. From this text I don't see that you take part in the experiment on animals. Of course I know that it is not quite true because you cited Ossowski et al. (32), but in this cited article there is no description about animals maintenance, there is no information about fattening performance of animals (daily gain or FCR or feed consumption - what is the most important for nutritional experiments). Where are these information? If you want to stay only with the description of the obtained laboratory results, focus on that and add that your study is a part of bigger experiment and so on... If you want to write something about the experiment on animals you have to add something more or write where these information readers can find: once again fattening performance, I didn't find any information obtained from so big number of animals - 1200 head, what's the average slaughtery weight, the length of fattening period ...
Where are the results of measurements of microclimatic condition (l.130- 131)?
What do you mean by "feed efficiency" (l.67)?
l.102 - you double "diet"
I think it will be better to introduce subsection discuss and present separately results and discuss. Presentation of your results is ok, but it will be better to discus results from both experiments together, especially because you assessed the same parameters in both experiments. You have to improve a little your discuss, sometimes it is rather introduction to your subject - perhaps it is worth to move some sentences to Introduction?, sometimes I felt unsatisfied from this part.
"Moreover, effects of supplementation (biochar vs. bentonite vs. zeolite) are highly variable and can be depend on the composition and dosage" - you had no too many samples.
I think that you self-cite too much and it is not always justified in the content of what you describe.
Author Response
Dear Reviewer,
The authors would like to warmly thank you for all comments and suggestions, especially the critical ones, aimed at improving the scientific value of the article and eliminating the most important errors. We greatly appreciate the opportunity that we have been given to further revise the manuscript. We believe that you will share the arguments submitted by authors and find this revision fully satisfactory.
Comments and Suggestions for Authors
Dear Authors,
generally your manuscript is not too bad but in my opinion you should rewrite it or just introduce some changes in the text.
First you have to decide what was the aim of your study. In the reviewed manuscript I found results' description of skeletal muscles and adipose tissues laboratory analyses - samples (n=35) were obtained from the nutritional (?) experiment and that's ok. From this text I don't see that you take part in the experiment on animals. Of course I know that it is not quite true because you cited Ossowski et al. (32), but in this cited article there is no description about animals maintenance, there is no information about fattening performance of animals (daily gain or FCR or feed consumption - what is the most important for nutritional experiments). Where are these information? If you want to stay only with the description of the obtained laboratory results, focus on that and add that your study is a part of bigger experiment and so on... If you want to write something about the experiment on animals you have to add something more or write where these information readers can find: once again fattening performance, I didn't find any information obtained from so big number of animals - 1200 head, what's the average slaughtery weight, the length of fattening period ...
Answer. Thank you for these key insights, which allow a more precise focus for our article. As suggested above, the title of the publication has been changed and information on the nature of this study has been supplemented, with the focus on the fatty acid profile of pork meat and fat as part of a larger whole on the effect of sorbents on the quality of raw materials obtained from pork carcasses.
Where are the results of measurements of microclimatic condition (l.130- 131)?
Answer. We would like to inform, that results of microclimatic condition measurements as well other analyses (analysis of biogenes in animal faeces, analysis of gas concentrations and content of total and respirable dust fractions) will be published in an upcoming issue of Przemysł Chemiczny (Ossowski M., Wlazło Ł., Bis-Wencel H., Krzaczek P., Nowakowicz-Dębek B. 2022. The use of natural sorbents in the diet of pigs as a method for reducing gaseous pollutants and manure nutrients from livestock housing. Prze. Chem. 101/5, 928-934. DOI: 10.15199/62.2022.5.XX)
What do you mean by "feed efficiency" (l.67)?
Answer. This part of text was redrafted and “feed conversion ratio’ was used.
l.102 - you double "diet"
Answer. Thank you for this remark, double word was deleted.
I think it will be better to introduce subsection discuss and present separately results and discuss. Presentation of your results is ok, but it will be better to discus results from both experiments together, especially because you assessed the same parameters in both experiments. You have to improve a little your discuss, sometimes it is rather introduction to your subject - perhaps it is worth to move some sentences to Introduction?, sometimes I felt unsatisfied from this part.
Answer. Thank you for this remark, authors have divided the content into results and discussion.
"Moreover, effects of supplementation (biochar vs. bentonite vs. zeolite) are highly variable and can be depend on the composition and dosage" - you had no too many samples.
Answer. Thank you for this suggestion, this disputable sentence was deleted.
I think that you self-cite too much and it is not always justified in the content of what you describe.
Answer. Thank you for this suggestion, authors have made every effort to avoid self-citation.
Reviewer 2 Report
Dear authors,
Many thanks for this piece of work. I found tour investigation is innovative and provide new insights.
However, beyond the scientific merit, I would expect the structure of the manuscript to be ameliorate In my opinion, you may improve the quality of writing and reporting facts.
For instance, very general comments are related to the need to clearly identify a gap in the literature. Another point is to report the legislation on sorbant use in formulations and authorization (at least in Europe) to use them. You may provide the reader with a background (not so detailed in the introduction, I would move those short paragraphs for each sorbant in the discussion when you provide the comments on the differences you pointed out) and rather focus on the mode of use and why you selected those different sorbant in comparative feeding, in few words the rationale of the investigation and the hypothesis you formulated. Thne, finally the aims. What did you expect ot find? Were the dosages tested following the mode of use, or previous experimental feeding?
Then, you must pay attention to english grammar and expressions. Sentences are sometimes too long and difficult to follow.
Please, provide more insights to comment on the link with data about fat.
Author Response
Dear Reviewer,
The authors would like to warmly thank you for all comments and suggestions, especially the critical ones, aimed at improving the scientific value of the article and eliminating the most important errors. We greatly appreciate the opportunity that we have been given to further revise the manuscript. We believe that you will share the arguments submitted by authors and find this revision fully satisfactory.
Comments and Suggestions for Authors
Dear authors,
Many thanks for this piece of work. I found tour investigation is innovative and provide new insights.
Answer. Thank you for this kind statement.
However, beyond the scientific merit, I would expect the structure of the manuscript to be ameliorate In my opinion, you may improve the quality of writing and reporting facts. For instance, very general comments are related to the need to clearly identify a gap in the literature. Another point is to report the legislation on sorbant use in formulations and authorization (at least in Europe) to use them. You may provide the reader with a background (not so detailed in the introduction, I would move those short paragraphs for each sorbant in the discussion when you provide the comments on the differences you pointed out) and rather focus on the mode of use and why you selected those different sorbant in comparative feeding, in few words the rationale of the investigation and the hypothesis you formulated. Thne, finally the aims. What did you expect ot find? Were the dosages tested following the mode of use, or previous experimental feeding?
Answer. Thank you for this suggestion, authors have made every effort to improve manuscript taking into account European legislations (EC, EFSA).
Then, you must pay attention to english grammar and expressions. Sentences are sometimes too long and difficult to follow.
Answer. Thank you for this suggestion, authors have made every effort to improve manuscript.
Please, provide more insights to comment on the link with data about fat.
Answer. As we mentioned, the literature on this topic is very sparse, so it is difficult to discuss our own results with those of other authors, at the same time the authors wanted to avoid speculation and limited themselves to basic facts.
Reviewer 3 Report
Report on the manuscript animals-1762996 entitled “The effect of natural sorbents in pig diets on the composition of fatty acids of muscle and adipose tissue”.
- Line 16-19. “…the efficiency of different clay minerals depends on chemical composition and physical features. To date, no studies have investigated the effect of the dietary addition of natural sorbents (biochar, bentonite-montmorillonite and/or zeolite-clinoptilolite) on the fatty acid content of pork (muscle and fat tissues).”
Please, carry out an English review of the whole manuscript.
- - The introduction is too long. This manuscript can be considered a continuation of another previously published by the same authors (Ossowski, M.; Wlazło, Ł.; Nowakowicz-Dębek, B.; Florek, M. Effect of Natural Sorbents in the Diet of Fattening Pigs on Meat Quality and Suitability for Processing. Animals 2021, 11, 2930. https://doi.org/10.3390/ani11102930). Therefore, there is no need for such a long Introduction:
o Lines 63-86. Can be shortened.
o Lines 87-94. What is the interest or the importance of this paragraph? It could be deleted.
- - Line 77. Are 4 references really necessary?
- - Line 86. Are 6 references really necessary?
- - Lines 127-128 and Tables S1 and S2. Please, describe in the text and as a note below the tables that both tables have been already published. They are a copy of Tables 1 and 2 of Ossowski et al. (2021).
- - Lines 152-164. It is a copy-paste from Domaradzki et al (2019). Please, delete or summarize.
- - Lines 167-172 (…of the carbon chain). Please, delete such a description/explanation. The nomenclature of the FAs is widely known.
- - Lines 173-180. More information or a better description is needed. Please, include in detail which FAs were considered for each FA indexes.
Which FAs were considered for TFA calculations? MUFA trans… meaning? 18:2 trans… meaning? 18:2 trans could be CLA (conjugated linoleic acid)?? CLA isomers must be described individually!!!
- - Lines 203-209. I do not agree with such comments. It has been widely described in the literature that MUFA or PUFA content can be modified by diet.
Did the authors analyze the FAs of the diet (feed)?
- - Line 205. Review the FA nomenclature.
- - Line 229. “…fatty acid composition…”
- - Lines 231-263. Only charcoal was discussed… what about the other considered components? Further discussion is needed!
- - Same comments for Tables 3 and 4… Further discussion is needed regarding the other dietary additives!
Why were AI and TI statistically significant for kidney fat but not for backfat?
Kidney fat uses? Implications?
Author Response
Dear Reviewer,
The authors would like to warmly thank you for all comments and suggestions, especially the critical ones, aimed at improving the scientific value of the article and eliminating the most important errors. We greatly appreciate the opportunity that we have been given to further revise the manuscript. We believe that you will share the arguments submitted by authors and find this revision fully satisfactory.
Comments and Suggestions for Authors
Report on the manuscript animals-1762996 entitled “The effect of natural sorbents in pig diets on the composition of fatty acids of muscle and adipose tissue”.
- Line 16-19. “…the efficiency of different clay minerals depends on chemical composition and physical features. To date, no studies have investigated the effect of the dietary addition of natural sorbents (biochar, bentonite-montmorillonite and/or zeolite-clinoptilolite) on the fatty acid content of pork (muscle and fat tissues).”
Answer. Thank you for this suggestion, this disputable sentence was deleted.
Please, carry out an English review of the whole manuscript.
Answer. Thank you for this suggestion, authors have made every effort to improve manuscript.
- The introduction is too long. This manuscript can be considered a continuation of another previously published by the same authors (Ossowski, M.; Wlazło, Ł.; Nowakowicz-Dębek, B.; Florek, M. Effect of Natural Sorbents in the Diet of Fattening Pigs on Meat Quality and Suitability for Processing. Animals 2021, 11, 2930. https://doi.org/10.3390/ani11102930). Therefore, there is no need for such a long Introduction:
o Lines 63-86. Can be shortened.
Answer. Thank you for this suggestion, authors have made every effort to improve the introduction.
o Lines 87-94. What is the interest or the importance of this paragraph? It could be deleted.
Answer. Thank you for this suggestion, this paragraph was deleted.
- Line 77. Are 4 references really necessary?
Answer. Thank you for this suggestion, two references were deleted.
- Line 86. Are 6 references really necessary?
Answer. Thank you for this suggestion, three references were left.
- Lines 127-128 and Tables S1 and S2. Please, describe in the text and as a note below the tables that both tables have been already published. They are a copy of Tables 1 and 2 of Ossowski et al. (2021).
Answer. Thank you for this valuable suggestion, appropriate information was introduced.
- Lines 152-164. It is a copy-paste from Domaradzki et al (2019). Please, delete or summarize.
Answer. Thank you for this suggestion, the description of the method was deleted.
- Lines 167-172 (…of the carbon chain). Please, delete such a description/explanation. The nomenclature of the FAs is widely known.
Answer. Thank you for this obvious remark, the description of FA was deleted.
- Lines 173-180. More information or a better description is needed. Please, include in detail which FAs were considered for each FA indexes.
Which FAs were considered for TFA calculations? MUFA trans… meaning? 18:2 trans… meaning? 18:2 trans could be CLA (conjugated linoleic acid)?? CLA isomers must be described individually!!!
Answer. Thank you for this remark, this part of text was improved.
- Lines 203-209. I do not agree with such comments. It has been widely described in the literature that MUFA or PUFA content can be modified by diet.
Did the authors analyze the FAs of the diet (feed)?
- Line 205. Review the FA nomenclature.
Answer. Thank you for your opinion, however, there is no mention of PUFAs in this quote from Wood et al. At the same time, given the mention in this section of trends found for PUFA, the authors have decided to remove this part.
- Line 229. “…fatty acid composition…” -
Answer. Thank you for this remark, this mistake was corrected.
- Lines 231-263. Only charcoal was discussed… what about the other considered components? Further discussion is needed! – because this is discussion of exp. I
Answer. Thank you for this remark, however, in Exp. I biochar (sorbent D) was tested, only.
- Same comments for Tables 3 and 4… Further discussion is needed regarding the other dietary additives!
Answer. Thank you for this suggestion, authors have reformatted this part of the article in line with the recommendations of other reviewers.
Why were AI and TI statistically significant for kidney fat but not for backfat?
Kidney fat uses? Implications?
Answer. This may be related to the different fatty acid profile resulting from the different location and function of these tissues. Pig fat is widely used in meat processing for the production of rendered fats and fritters.
Round 2
Reviewer 1 Report
Dear Authors,
I like improved version of your manuscript but still I don't like the discuss.
First correct sentences (l.99-101), remove: "The body weight
99 of pigs was between 105.4 and 108.6 kg." and add this information here: "Briefly, 6 fatteners (slaughter weight between 105-108 kg) from each group...
From the Discusss delate or remove to Intorduction all sentences from lines: 221-237 and 276-285.
In my opinion after these small changes the manuscript could be printed.
Author Response
Dear Reviewer 1,
The authors would like to warmly thank you once again for all comments and suggestions, as well as time put into the review of the manuscript. We believe that you will find this revision fully satisfactory.
Dear Authors,
I like improved version of your manuscript …
Answer: Thank you very much for kind statement.
but still I don't like the discuss.
First correct sentences (l.99-101), remove: "The body weight
99 of pigs was between 105.4 and 108.6 kg." and add this information here: "Briefly, 6 fatteners (slaughter weight between 105-108 kg) from each group...
Answer: Thank you for this tip. The sentence was redrafted.
From the Discusss delate or remove to Introduction all sentences from lines: 221-237 and 276-285.
In my opinion after these small changes the manuscript could be printed.
Answer: Thank you for this suggestion. Authors finally decided to remove the indicated sentences, as in the previous review, Reviewer 3 suggested limiting the information contained in the Introduction and its significant shortening.
Reviewer 3 Report
Report on the manuscript animals-1762996R1.
The authors have improved the manuscript considerably.
Nevertheless, some other comments:
- Please, review again the nomenclature of the fatty acids. For instance, Line120, C14:1c9 = C14:1n-5, C15:1c9 = C15:1n-6, C16:1c7 = C16:1n-9, C16:1c9 = C16:1n-7, C17:1c9 = C17:1n-8, C18:1c9 = C18:1n-9, C18:1c11 = C18:1n-7, etc… (along the whole manuscript and table!)
- I would remove ∑TFA from the whole manuscript. It is not statistically significant and the presence of trans fatty acids could be related to the methylation process not to the animal metabolism.
Author Response
Dear Reviewer 3,
The authors would like to warmly thank you once again for all comments and suggestions, as well as time put into the review of the manuscript. We believe that you will find this revision fully satisfactory.
The authors have improved the manuscript considerably.
Answer: Thank you very much for kind statement.
Nevertheless, some other comments:
- Please, review again the nomenclature of the fatty acids. For instance, Line120, C14:1c9 = C14:1n-5, C15:1c9 = C15:1n-6, C16:1c7 = C16:1n-9, C16:1c9 = C16:1n-7, C17:1c9 = C17:1n-8, C18:1c9 = C18:1n-9, C18:1c11 = C18:1n-7, etc… (along the whole manuscript and table!)
Answer: Thank you for this suggestion. The fatty acid nomenclature has been revised throughout the manuscript and tables and amended as recommended.
- I would remove ∑TFA from the whole manuscript. It is not statistically significant and the presence of trans fatty acids could be related to the methylation process not to the animal metabolism.
Answer: Although, in the literature, it is indicated that even in monogastric animals, biohydrogenation of unsaturated fatty acids and the formation of traces of trans FAs may occur, you may be right that they may also result from the methylation process. Therefore, according to the suggestion, it was decided to remove this group of FA from manuscript.